# Reading Comprehension: An Essential Process for the Development of Critical Thinking

Narcisa Medranda-Morales [1,*] , Victoria Dalila Palacios Mieles [2] and Marco Villalba Guevara [3]

1   Communication College, Universidad Politécnica Salesiana, 170517 Quito, Ecuador
2   Faculty of Educational Sciences, Pontificia Universidad Católica del Ecuador, 170525 Quito, Ecuador; vdpalacios@puce.edu.ec
3   Education College, Universidad Politécnica Salesiana, 170517 Quito, Ecuador
*   Correspondence: nmedranda@ups.edu.ec

**Abstract:** People have the right to receive a quality of education that promotes, among other things, reading comprehension and the development of critical thinking. In Ecuador, there is a need to implement strategies and design educational programs to encourage and facilitate the development of these skills. This research focuses on the relationship between reading comprehension and the development of critical thinking, with a particular interest in the education of high school students in Ecuador. This research aims to analyze the relationship between reading comprehension and critical thinking, based on the experiences of a sample of high school students of both genders, aged between 12 and 16 years, from five schools in different regions of Ecuador. The study adopts a quantitative, retrospective, and correlational approach. The surveys, administered to a sample of students, showed that teachers actively encourage reading and implement timely educational strategies and teaching methods to develop critical thinking, recognizing its crucial role in future professional performance.

**Keywords:** reading comprehension; critical thinking; high school education

## 1. Introduction

This research analyzes the relationship between reading comprehension and the development of critical thinking, focusing on the education of high school students in Ecuador. Both reading comprehension and critical thinking are complex cognitive processes. To understand their relationship, it is essential first to analyze one of the basic communicative skills: reading.

There are several definitions of reading, but it is generally described as an exclusively human ability that involves biological, psychosocial, and affective processes leading readers to establish relationships between what they read and their experiences, thus strengthening their capacity for analysis and argumentation [1].

Reading is the ability to decode the information implicit in oral or written texts. Different reading levels are developed through formal, nonformal, or informal education.

The primary level requires basic skills such as decoding, grapheme identification, grapheme–phoneme conversion, word recognition, and syntactic-function assignment [2,3].

At the literal level, the reader focuses on the ideas and information explicitly stated. In narrative texts, the reader identifies sequences of actions, characters, and scenarios; in expository or argumentative texts, the reader identifies primary and secondary ideas, comparisons, exemplifications, and summarizes, synthesizes, and creates synoptic tables and concept maps [3,4].

The reader demonstrates high-level skills such as abstraction, comprehension, analysis, and interpretation at the inferential level. The reader relates meanings; adds information, experiences, knowledge, and prior ideas; reads between the lines and de-duces implicit information; hypothesizes and generates new ideas; organizes and ex-plains information; appropriates meaning; and draws conclusions [2–4].

At the critical level, the reader makes value judgments and evaluates the content read and the knowledge acquired [3].

Three models describe how reading occurs. The top-down model states that the information stored in memory controls the reading process. The bottom-up model establishes that the essential part of the reading process is the linear and automatic decoding of textual information. The interactive model states that reading comprehension is the product of interactions between the reader's prior knowledge and the information in the text [5].

Reading comprehension is fully achieved when the reader goes beyond the literal level and reaches at least the inferential level. Understanding involves approaching different text types, grasping the author's intention, having a reading goal [6], perceiving the text structure, and self-regulation [7]. Acquiring new knowledge, ideas, and judgments from reading comprehension leads to personal development [2]. This is why developing the skills and abilities to form high-level readers is essential. According to [8], whenever one reads, one thinks, refines criteria, contrasts and questions ideas, and learns to learn, think, and enjoy.

Some factors may hinder the attainment of comprehensive reading levels. First, poor reading habits make constant and sustained practice difficult. Second, deficiencies in the basic decoding level cause the reader to concentrate on letters and words, not meanings. The lack of precise objectives to understand the task, and the lack of a vocabulary and a conceptual network to help relate and understand the information in the text, are also factors that affect the development of reading comprehension [9].

The reading comprehension process is also influenced by: memory problems that prevent the information and knowledge obtained through reading from lasting; low self-esteem; a lack of interest and motivation; and the excessive use of technology that diverts people's attention to social networks, taking them away from texts that require more complex comprehension skills [9].

The absence of comprehensive reading leads to functional illiteracy, and diminishes the opportunities to acquire learning and experiences conducive to personal development. It is important to consider that the lack of strategies to promote and enhance reading limits the potential for developing critical thinking [10], due to the close relationship between them.

John Dewey used reflective thinking at the beginning of the 20th century. Later, Robert Ennis and Peter Facione delved deeper into the cognitive processes of thought and referred to critical thinking. Both scholars [11,12] tried to define and characterize it, establishing an ideal critical thinker's cognitive skills and motivational dispositions. To summarize, critical thinking refers to reflection and reasoning in order to make decisions and solve problems based on knowledge of facts.

Critical thinking skills are interpretation, analysis, evaluation, inference, explanation, assessment, integration, and self-regulation. Critical thinkers identify essentials, analyze arguments, ask questions, identify assumptions, search, integrate, and organize information. They can convey meaning, explain, justify facts, offer logical reasons, seek alternatives, and use their skills and resources. They are curious, reasonable, systematic, self-confident, self-aware, empathetic, sensitive, and sympathetic. Curiosity, understanding, open-mindedness, reasonableness, honesty, prudence, impartiality, and accuracy are some of the factors that lead critical thinkers to develop these skills.

Critical thinking is a complex knowledge process that requires the individual to develop logical, critical, and pragmatic skills to search for truth based on knowledge, facts, and evidence and make a personal judgment and assessment [13–16].

Reading comprehension is the starting point for developing critical thinking [17]. Both are cognitive macro-processes that require skills for processing and producing information and knowledge [5]. The critical level is the highest and most complex level of reading comprehension, because it involves evaluating and assessing the information in the text after going through a process of analysis, inference, interpretation, and explanation. To achieve this level of reading comprehension, evaluation, and judgment, it is essential to have specific critical thinking skills.

Curiosity, open-mindedness, self-confidence, and self-awareness are motivational conditions characteristic of the critical thinker that favour inferential and critical reading, that is, reading comprehension. The association of meanings, formulation of hypotheses, generation of new ideas, elaboration of conclusions, and evaluation are essential for making decisions, solving problems, and developing critical thinking.

There is an evident relationship between reading comprehension and critical thinking that cannot be overlooked. Developing critical thinking involves reflection, analysis, and evaluation of information. Complex reading strategies that encourage inference, connecting information with prior knowledge, organizing and interpreting the information correctly, and appropriating meaning are needed to achieve this.

Reading comprehension involves deep analysis to understand better the objective reality and the environment in which the individual develops, significantly influencing the development of critical thinking.

Autonomous learning and knowledge acquisition are based on reading comprehension. However, the individual is not born with this ability, which only develops with effort, time, and dedication [6].

Most readers need in-depth instruction and high quality education to achieve high reading comprehension levels. Because of this, educational systems must contribute to developing reading skills by implementing strategies and providing appropriate resources to motivate students and foster reading habits [2,7].

Society needs people who can analyze and assess the environment to make decisions and solve problems efficiently. In [18], the reflective paradigm in critical academic practice is explored, stating that one of the goals of education is to train people be capable of analyzing and understanding the world around them. The most important thing is not the acquisition of information but the search for existing relationships. Teachers should guide this search and promote the juxtaposition of the knowledge acquired in the different subjects.

Educational systems have the responsibility to address in a transversal way the development of critical thinking [19], to develop skills through educational processes of multiple dimensions that encourage observation, assessment, questioning of reality, and through strategies to analyze information, make decisions, and solve problems [20]. Programs that include didactic strategies to develop reading comprehension skills and critical thinking are valuable for educational systems designed to form autonomous, reflective, and conscious citizens of changes and personal and social development.

Some authors [21,22] claim that the Regional Reading Center for Latin America and the Caribbean (CERLAC-UNESCO) recorded in 2012 that each person reads 0.5 books per year in Ecuador, as opposed to Chile and Argentina, where each person reads 5.4 and 4.6 books per year, respectively. However, in [23], the National Plan for the Promotion of Books and Reading author states that CERLAC denied disseminating this figure.

Figures from the National Institute of Statistics and Census (INEC) [24] indicate that 27% of Ecuadorians do not have the habit of reading, while 73.5% do. Of those who do not have reading habits, 56.8% do not read due to a lack of interest. Another essential datum provided by INEC is that 50.3% of Ecuadorians read for 1 or 2 h per week.

In 2021, the Ministry of Culture and Heritage of Ecuador conducted the Survey of reading habits, practices, and cultural consumption [25] to evaluate, among other aspects, the reading habits of Ecuadorians. As a result, it was found that 91.4% of the Ecuadorian population knows how to read, and 76.7% of people read daily in print or digital format. The most used devices for reading are the cellphone (56.7%) and printed material (33.9%). The survey also revealed that in Ecuador people read one book completely and two books partially per year.

Regarding students, the National Institute for Educational Evaluation (Ineval) presented in 2018 the results of the OECD's Program for International Student Assessment (PISA), applied in Ecuador in October 2017 to more than 6100 students from 3rd grade to 8th grade of high school. Concerning reading, the results showed that Ecuadorian students were at the average (406 points) for Latin American and Caribbean countries (408 points).

Ecuadorian students were in a lower position than the OECD average for the rest of the world (493 points) [26].

The National Institute for Educational Evaluation (Ineval) (2022) recently presented the results obtained in the process "SER Estudiante 2021–2022". A total of 22 thousand students from 690 educational institutions were evaluated, showing that the grades in Language and Literature were lower than those of the previous year [27].

The data above showed that Ecuador requires public policies to encourage people's interest in reading. As for the student population, it is necessary to design educational programs to improve students' performance in Language and Literature and develop reading comprehension skills to develop critical thinking.

This study aimed to provide information on developing reading comprehension and critical thinking skills processes in the Ecuadorian educational system. This information is helpful for teachers who teach classes and those in charge of designing pedagogical action programs. Didactic intervention plans should be based on knowledge generated by rigorous studies that provide reliable data.

In recent years, several investigations have been conducted on reading comprehension, developing critical thinking, and their relationship. In [7] it is mentioned that readers need a process of formation of reading skills and strategies to improve comprehension. This is even more necessary for students with difficulty inferring and self-regulating understanding. That was why they set out to evaluate the effectiveness of an intervention program for teaching high-level skills. The program was applied to 4th-grade children from middle socioeconomic level schools in the Autonomous City of Buenos Aires and Gran Buenos Aires in Argentina. The groups to which the program was applied significantly improved their reading comprehension skills.

In [2], the reading comprehension level of Chilean students from 2nd to 8th grade in a subsidized urban educational institution was investigated. The differences among students were analyzed, and the main reading difficulties were identified to find solutions. They found that reading comprehension decreased as students moved up to higher grades. It was also found that self-regulation strategies for the reading process need to be improved and that there were no significant differences between boys and girls. They suggested the implementation of different approaches to enhance students' reading comprehension.

In [6], the Reading Proficiency Test for High School Education was applied to 13- and 14-year-old students in 9th grade at Julio Pérez Ferrero Educational Institution in Cú-cuta, Colombia, to evaluate the reading comprehension level and its relationship to academic performance. The results showed that the students who took the test had textual, inferential, and contextual difficulties. The correlational analysis of the data from the different subjects showed a significant positive relationship, although this does not necessarily imply a cause-and-effect association.

In [3], a study was conducted to determine how reading comprehension influenced school performance, taking as a sample the 7th-grade students of "Manuelita Sáenz" Educational Institution in Cotopaxi, Ecuador. With a registration test, all the Language and Literature classes were observed. It was determined that the factors that most affected reading comprehension are the lack of reading habits, level of interest in the topics addressed, and inadequate reading techniques. It was concluded that there is a close relationship between reading comprehension and school performance.

In [22], the reading comprehension of bilingual students in 3rd grade of Elementary Education from five rural intercultural educational institutions in Colta, Ecuador, was evaluated. The study demonstrated the relationship between personality variables and reading comprehension level. It was concluded that most students assessed did not have a high reading comprehension level and that educational and socio-demographic aspects influenced these results.

In [28], a study was conducted at Francisco José Caldas Educational Institution in Soledad-Atlántico, Colombia. Collaborative work was analyzed as a didactic strategy to develop critical thinking in 5th-grade students. Teachers and students were interviewed. It

was shown that teachers are not familiar with collaborative work, so teachers must learn to apply this strategy to determine if students learn and develop critical thinking by doing collaborative work.

In [20], didactic strategies to develop critical thinking implemented in communication with high school students in educational institutions in Ocros, Peru were evaluated. Workshops were conducted, and information was collected through the Metaplan technique. The results showed that students needed the cognitive tools to develop critical thinking, and teachers must implement strategies to build these skills. This is why reflecting on educational policies, teacher training processes, and curriculum design is necessary.

In [29], the low reading comprehension level of middle school students of Los Vergeles Educational Institution, in Guayaquil, Ecuador, in 2017 was addressed. The objective was to determine the incidence of reading comprehension in developing critical thinking and to elaborate a methodological guide that developed students' skills. Through the application of standardized tests and interviews, it was found that students did not express their ideas clearly or had not developed critical thinking. Guidance activities related to the importance of reading comprehension for developing critical thinking were proposed.

In [17], research was conducted to implement didactic strategies to develop critical thinking by improving the reading comprehension of 10th-grade students of San Viator High School in Tunja, Colombia. The interviews, exploration workshops, and surveys showed that students must develop their reading comprehension skills to develop critical thinking. Considering the research results, it was proposed to implement strategies to improve inquiry and reasoning skills.

In [30], the reading comprehension and the development of critical thinking of 1st-grade BGU students of Aníbal Salgado Ruiz Educational Institution in Tisaleo, Ecuador, were analyzed. The objective was to collect baseline and numerical information to evaluate students' reading comprehension and critical thinking to determine the relationship between them. The results showed that reading comprehension significantly influenced the development of critical thinking, so both variables should be addressed and improved in the teaching–learning processes.

In [18], a quasi-experimental study with a control group was conducted to determine the global and differential effects of the programmatic development of critical thinking skills on the reading comprehension of 57 8th-grade students from subsidized schools in Marga-Marga, Valparaíso, Chile. The results showed that the programmatic development of critical thinking skills significantly improved reading comprehension and textual processing.

This research aimed to analyze the relationship between reading comprehension and critical thinking through a study conducted with high school students of Marcos Benetazzo Educational Institution. The reference provided by Referee 1 to be cited in this section is not available.

## 2. Materials and Methods

This study has adopted a quantitative approach with a correlational design. The main objective is to analyze the relationship between reading comprehension (independent variable) and the development of critical thinking (dependent variable) in high school students in Ecuador.

The study participants were selected through a stratified random sampling method applied in five schools from different regions of Ecuador. As a result, 200 students between the ages of 12 and 16 were selected.

The Progressive Complexity Reading Comprehension Test (PCRCT) was used to assess the participants' reading comprehension. This instrument had demonstrated high validity and reliability in previous studies.

The Watson–Glaser Critical Thinking Scale was used to measure the critical thinking variable. This scale has been validated in multiple studies and has a Cronbach's alpha of 0.85.

The participating students completed both instruments in a controlled environment and under the supervision of the researchers. Confidentiality and anonymity of the collected data were ensured.

The collected data were analyzed using the statistical software Statistical Product and Service Solutions (IBM SPSS Statistics 21.0 (2012)). Pearson correlation tests were employed to assess the relationship between variables. Furthermore, multiple regression analyses were conducted to control for potential confounding variables.

All participants and their legal guardians signed an informed consent form before participating in the study. The Ethics Committee of the Universidad Politécnica Salesiana approved the study.

## 3. Results and Discussion

Table 1 displays the reliability and normality measures for each variable and dimension. All reliability values fall within the range of 0.7 to 0.9, indicating a high internal consistency of the questionnaire. Additionally, the *p* values for the normality of each variable are less than or equal to 0.05, suggesting that the data follow a normal distribution. This level of reliability supports the robustness of the measurement instrument and the validity of the results obtained.

**Table 1.** Analysis of the reliability of the questionnaire and normality of the data.

| Variable | Reliability | Normality | Sig. |
| --- | --- | --- | --- |
| VT. Reading comprehension | 0.786 | 0.188 | |
| DT. Global reading | 0.812 | | |
| DT. Problem solving | 0.791 | | |
| DT. Reading support | 0.698 | | |
| VT. Critical thinking | 0.746 | 0.199 | 0.000 |

Source: own elaboration.

Table 2 displays the correlation between reading comprehension and critical thinking. A significant correlation ($p < 0.05$) with a correlation coefficient of 0.871 was found. This value demonstrates a strong positive linear relationship between both variables, indicating that the development of reading comprehension could significantly impact the development of critical thinking.

**Table 2.** Analysis of the correlation obtained through the study of the variables: students' reading comprehension and critical thinking.

| | Reading Comprehension | Critical Thinking | Sig. (Bilateral) | N |
| --- | --- | --- | --- | --- |
| Reading comprehension | 1.000 | 0.871 | | 345 |
| Critical thinking | 0.871 | 1.000 | 0.000345 | |

Source: own elaboration.

Figure 1 shows that 88.4% of the participants agreed that teachers motivate them to read, which can positively impact the development of critical thinking.

Figure 2 illustrates that 81.16% of the participants believe that developing reading habits is essential for their professional development. This represents a significant incentive to engage in tasks that enhance reading comprehension skills and develop critical thinking.

Figure 3 depicts that 52.17% of the participants believed teachers' educational actions regarding reading habits were timely and effective, while 42.03% disagreed. Considering and analyzing this result is important, as teachers' actions are crucial for fostering meaningful and sustainable learning.

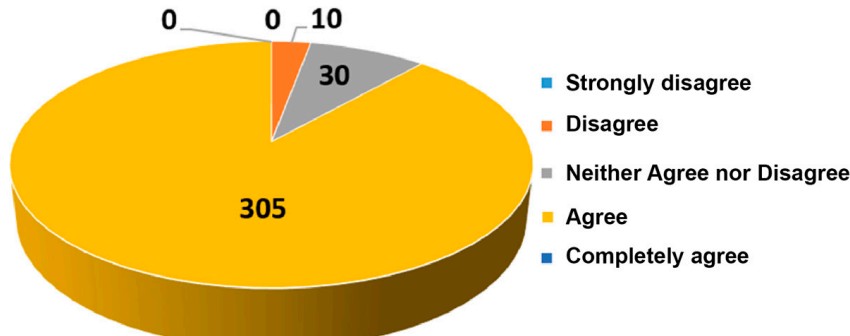

**Figure 1.** Answers to question 1. Teachers motivate students to read, which favours the development of critical thinking.

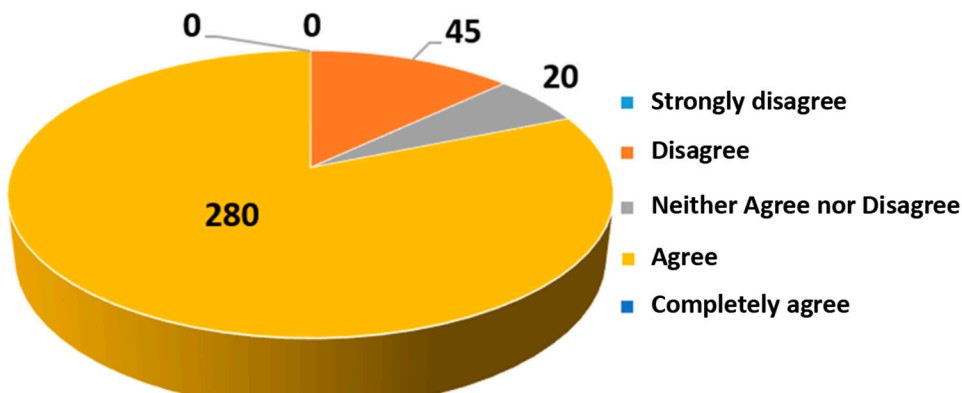

**Figure 2.** Answers to question 2. Reading habits are essential for the development of professional skills in students.

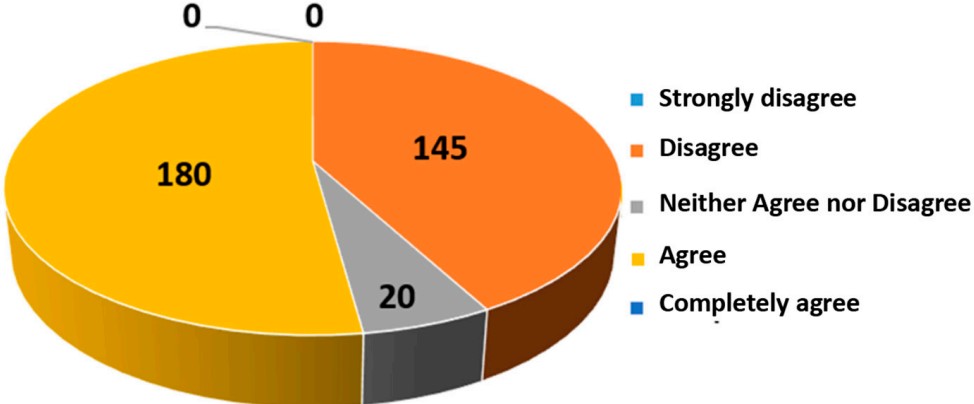

**Figure 3.** Answers to question 3. Teachers' educational actions regarding reading habits were timely and effective.

Figure 4 illustrates that 76.81% of the respondents agreed with the relevance and appropriateness of the teaching methods and their contribution to improving students' critical thinking. The relevance and suitability of the methods contribute to raising the quality of education and achieving meaningful learning, leading to the development of critical thinking.

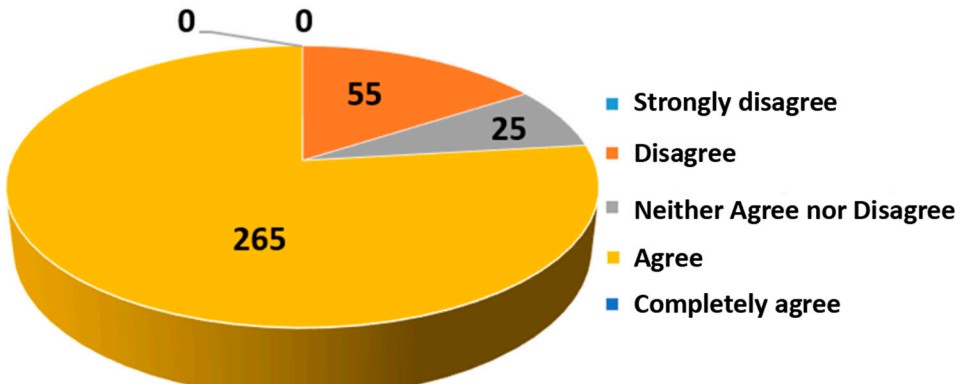

**Figure 4.** Answers to question 4. Teaching methods were relevant and appropriate, which contributed to the development of critical thinking and improved student learning.

The reliability indicators, with values ranging from 0.7 to 0.9, demonstrate that the instruments used in this study are highly reliable. This ensures that the results are consistent and representative of the studied population.

The correlation coefficient of 0.871 demonstrates a strong positive linear relationship between reading comprehension and the development of critical thinking in high school students in Ecuador. This finding supports previous investigations, which have identified a positive correlation between these two variables [30].

Most students (88.4%) believed their teachers motivated them to read. This is fundamental to developing critical thinking since motivation is a crucial driver for learning and acquiring complex cognitive skills.

Of the participants, 81.16% believed reading habits are essential for professional development. This perception can encourage educators to implement strategies and incorporate more reading activities into the curriculum.

The results of this study have a significant impact on the educational system in Ecuador. Educators can use these findings to design more effective interventions to enhance students' reading comprehension and critical thinking. It is advisable to design and implement didactic interventions that address the issue of reading comprehension. This proposal is based on specific studies that have demonstrated that students require a training process in reading skills and strategies to improve comprehension [2], as they face difficulties in textual organization, inference, and context [6], Additionally, students do not express themselves clearly or develop ideas [29] and lack the tools to develop critical thinking [20].

Systematic programs can develop students' critical thinking, reading comprehension, and textual processing [18]. It is recommended to design tools for fostering inquiry and reasoning [17]. It has been proven that an intervention program for the explicit teaching of complex skills significantly improves reading comprehension [7] and promotes the development of students' critical thinking.

## 4. Conclusions

The correlation between reading comprehension and the development of critical thinking has been studied by several researchers, which was demonstrated through the documentary analysis conducted in this research. The findings of this research corroborate the existence of a statistically significant, substantial, and positive linear correlation between the two variables. The questionnaires applied to high school students of Marcos Benetazzo Educational Institution allowed and favoured the study of this phenomenon.

Due to the existing correlation and the undoubted importance of developing critical thinking, it is recommended that educational systems design and implement educational policies and pedagogical programs to promote reading and improve reading comprehension from different perspectives and levels of complexity. This would, in turn, improve the professional skills of high school students.

This research can provide valuable and innovative insights for future studies. The findings from the surveyed students regarding their educational environment and the development of reading comprehension and critical thinking highlight the importance of designing and implementing training programs for teachers and implementing suitable didactic interventions for high school students in Ecuador.

Selecting a larger and more diverse population sample in future research is also advisable. Including various educational institutions to obtain and analyze comprehensive results would be beneficial. By doing so, efficient programs can be designed and developed to foster the development of reading comprehension and critical thinking skills.

Finally, it is worth mentioning that a theoretical and methodological limitation was identified during this research. This limitation stemmed from the lack of didactic interventions focused on reading comprehension and critical thinking in Ecuador. To overcome this limitation, a thorough review of studies conducted in other South American countries was carried out, as educational issues in the region share similarities despite the evident differences. These studies provided valuable insights for conducting preliminary observations that were crucial for program design.

**Author Contributions:** Conceptualization, N.M.-M.; Methodology, N.M.-M. and V.D.P.M.; Validation, M.V.G.; Investigation, M.V.G.; Resources, N.M.-M.; Writing—review & editing, N.M.-M. and V.D.P.M.; Visualization, M.V.G. All authors have read and agreed to the published version of the manuscript.

**Funding:** This research received no external funding.

**Institutional Review Board Statement:** The study was conducted in accordance with the Declaration of Helsinki, and approved by the Institutional Review Board of Salesian Polytechnic University (protocol code 0525-021-2023-07-04 and 12 April 2023).

**Informed Consent Statement:** Informed consent was obtained from all subjects involved in the study.

**Data Availability Statement:** The data presented in this study are available on request from the corresponding author. The data are not publicly available due to privacy for students.

**Conflicts of Interest:** The authors declare no conflict of interest.

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
