# Peer review of "Reading Comprehension: An Essential Process for the Development of Critical Thinking"

_education, doi:10.3390/educsci13111068_

Round 1
Reviewer 1 Report
The paper titled "Reading comprehension: an essential process for the development of critical thinking" is reviewed. A number of related recommendations are presented as follows:
· In the introduction part, it may be better to make general definitions instead of the study purpose.
· The authors should first define abbreviations before using them
· In addition, The authors might use an abbreviation table at the beginning or end of the text since lots of abbreviations are used in the text.
· It is seen that the authors wrote the literature in great detail and emphasized the research purpose more than once. Instead, it may be better to include additional descriptive information in the method part.
· In addition, authors should explain about the questionnaire. Developed by whom etc.
· The authors should add around 100 more words to the conclusion section of your report. Include the future scope of your research study, limitations encountered during the study, and procedures to overcome them.
Author Response
Dear reviewer:
In the attached file I have included the observations made in the text, the same ones that have allowed the article to be significantly enriched.
I have attached the article for your review.
Cordially

Reviewer 2 Report
It is a very interesting and suitable work to be published, but some considerations must be taken into account. Specific details are highlighted and commented on in the attached document. However, I do believe that the authors should consider redrafting the section on material and method and the results. The material and method section must be very clear about the characteristics of the participating students. Regarding the different instruments, I think they should be explained in detail, in particular, how to calculate the global scores, since later it is not understood how the correlations are obtained in the results section. These are powerful instruments, with a large sample, however the results are scarce, much more information could be obtained. I think that the correlations are not correct when we talk about Likert scale questionnaires. These are ordinal variables, and therefore relative statistics must be applied to ordinal categorical variables. Obtaining probabilities. Likewise, if a relationship between variables is observed, could one be determined through the other? I think they should review the comments and try to improve these two sections

Author Response

(The authors gave the same response as above.)

Reviewer 3 Report
First, I appreciate the author(s) line of research and found it to be informative and useful, especially to High Schools and educators in Ecuador. I feel like there is definitely room in the extant research for studies such as this and that it helps provide further data to motivate students to want to learn and be in control of their learning, the importance of comprehension and the link to critical thinking and the need for educators to have sustained professional development to help learners make these connections.
Beginning on line 56, the authors may want to consider expanding factors that hinder attainment of comprehension. Some of those hinderances could be fluency which is related to deficiencies in decoding, but could also reading prosody. Also, I think discussing the role of vocabulary acquisition should also be mentioned. Before entering kindergarten, the expectation is that learners have around 5,000 to 10,000 words in their oral vocabularies. However, learners from economically disadvantaged backgrounds or non-native speaking learners may have that number cut in half and thus have a difficult time catching up with their more literacy advantaged peers and often struggle with comprehension.
Short disjointed choppy paragraphs. Look to combine and synthesize paragraphs. (104-113)
200- Rewrite first sentence. Same for 206. They currently appear to be incomplete sentences. Regardless they need to be rewritten. Perhaps the author(s) should consider 153-240 as all of these are disjointed paragraphs focused on one study at a time. Work toward a synthesis of ideas rather than a choppy series of studies. To be a little picky, I found the questions and survey tables/figures to be difficult to read as they were centered with the table. I feel it would be easier to follow if they were left justified.
The quality of English was strong, however some of the paragraphs seemed choppy and disjointed. I would encourage the author(s) to work on a synthesis of ideas as opposed to reporting on single study after study.
Author Response
Good afternoon:
I have incorporated the recommendations that he made to me in the text, attached for your review.
